# Practical Recommendations for Maintaining Active Lifestyle during the COVID-19 Pandemic: A Systematic Literature Review

**DOI:** 10.3390/ijerph17176265

**Published:** 2020-08-28

**Authors:** Ellen Bentlage, Achraf Ammar, Daniella How, Mona Ahmed, Khaled Trabelsi, Hamdi Chtourou, Michael Brach

**Affiliations:** 1Institute of Sport and Exercise Sciences, University of Münster, 48149 Münster, Germany; d.how@uni-muenster.de (D.H.); mona.ahmad@uni-muenster.de (M.A.); michael.brach@uni-muenster.de (M.B.); 2Institute of Sport Sciences, Otto-von-Guericke University, 39104 Magdeburg, Germany; achraf1.ammar@ovgu.de; 3High Institute of Sport and Physical Education of Sfax, University of Sfax, Sfax 3000, Tunisia; trabelsikhaled@gmail.com (K.T.); h_chtourou@yahoo.fr (H.C.); 4Research Laboratory: Education, Motricité, Sport et Santé, EM2S, LR19JS01, Sfax 3000, Tunisia; 5Research Unit: Physical Activity, Sport, and Health, UR18JS01, National Observatory of Sport, Tunis 1003, Tunisia

**Keywords:** COVID-19, pandemic, social isolation, quarantine, physical activity, recommendations

## Abstract

Diminished volumes of habitual physical activity and increased sedentary levels have been observed as a result of COVID-19 home-confinement. Consequences of inactivity, including a higher mortality rate and poorer general health and fitness, have been reported. This systematic review aimed to provide practical recommendations for maintaining active lifestyles during pandemics. In May 2020, two electronic databases (PubMed; Web of Science) were used to search for relevant studies. A total of 1206 records were screened by two researchers. Thirty-one relevant studies were included in this systematic review, in which the methodological quality was assessed. With regard to six studies, which explicitly dealt with physical activity during COVID-19, the evidence level is classified by three articles to level II, and in the other three to level VI. Regarding the physical activity recommendations in these papers, three of them were classified to a medium, and the same number to a weak evidence base. Of the 25 papers which refer to other pandemics and/or isolation situations, one was classified to evidence level I, four were ranged to level II, three to level III, one to level V, and the others to level VI. This systematic review revealed that reduced physical activity levels are of serious concern during home confinement in pandemic times. The recommendations provided by many international organizations to maintain active lifestyles during these times mainly target the general population, with less consideration for vulnerable populations (e.g., older adults, people with health issues). Therefore, personalized and supervised physical activity programs are urgently needed, with the option to group-play physical activity programs (e.g., exergames). These can be assisted, delivered, and disseminated worldwide through information and communication technology solutions. If it is permitted and safe, being active outside in daylight is advised, with an effort level of mild to moderate using the rating of perceived exertion scale. Relaxation techniques should be integrated into the daily routine to reduce stress levels. On the evidence base and levels of the included articles in this review, the results need to be interpreted with caution. Given that policies are different across regions and countries, further research is needed to categorize recommendations according to different social-distancing scenarios.

## 1. Introduction

COVID-19 was first detected in late 2019 in China, and has since spread throughout the world in a short period to affect more than 10 million people (28 June 2020), including nearly half a million deaths [1]. On 30 January 2020, the World Health Organization (WHO) declared the outbreak of COVID-19 disease as a global health emergency and on 11 March, the disease was proclaimed a global pandemic [2]. Due to the ever-growing number of confirmed cases, and to avoid overwhelming health systems, lockdowns with home-confinement and social distancing were implemented [3]. Practically, this meant gyms, public parks, sports grounds, outdoor playing areas, and schools were closed, among other facilities [4]. This situation has negatively affected children and youth, who normally have an active lifestyle through active transport, school-related activities, and sport participation [5]. The resulting reduction in social participation and physical activity (PA) during this home confinement is of serious concern for older adults too, as they are typically more inactive compared to younger aged individuals and more prone to chronic diseases [6]. A quantitative measurement of one-week’s average steps for 30 million users of the Fitbit© device, comparing March 2020 to March 2019, showed a fundamental reduction in average steps in almost all countries [7].

Research shows that physical inactivity is a high-risk factor for major disease morbidity [8]. A large-scale, prospective study demonstrated that 30 min per day of light-intensity PA, which replaces sitting for the same duration, is able to reduce the mortality risk from cardiovascular disease by 24% [9]. According to the WHO, regular PA has multiple health benefits as evidenced by reducing high blood pressure, weight management, reducing the risk of type 2 diabetes, stroke, heart disease, and various cancers, which are all variables that can increase susceptibility to the Acute Respiratory Infection (ARI) pandemic [10]. Importantly, a recent systematic review reported that exercise reduced the severity of ARI symptoms among 373 participants and the number of symptom days (−2.24 days) during the follow-up period measured on the Wisconsin Upper Respiratory Symptom Survey (WURSS-24) [11].

With recent predictions indicating that the COVID-19 pandemic could last over 18 months [12], the negative impact on people’s lifestyle with the low level of PA and the increased sedentary behavior could persist, with a resulting serious and detrimental effect on health. Therefore, it is essential to promote activity under confinement conditions. In order to develop the scientific evidence for appropriate recommendations and interventions, our group implemented a research program with three interacting branches: The survey branch addresses the behavioral and lifestyle consequences of COVID-19 restrictions of the population during different pandemic stages [13,14,15].The main findings from Europe, Western Asia, North-Africa, and America showed that the COVID-19 home confinement had a negative effect on social activity [15] and all PA intensity levels [13,14], with the number of days per week of PA lowered by 22.7%, 24%, 35%, respectively, for vigorous intensity, moderate intensity, and walking. Additionally, the total weekly PA (MET-min·week-1) was 38% lower during home confinement, while daily sitting time increased from 5 to 8 h per day [13,14] Subgroup analyses addressing different target groups and confinement levels will follow.The review branch started with an early review on the effects of quarantine and social distancing on well-being and other psychological aspects, which summarized possible recommendations for staying physically active as a mitigation of these effects and risks [16]. The present systematic review aims to identify, evaluate, and summarize existent practical recommendations to promote PA during the actual pandemic or any other isolation period.The intervention branch will build on the published results of the other branches and will in return contribute to them. It is in the grant application phase and will be reported elsewhere.

The present study belongs to the review branch of the research program. It builds on the evidence for low PA, which has been investigated in the survey branch, and should advance the development of specific recommendations in the intervention branch. Consequently, the research question of this review is: “Which practical recommendations to maintain or to promote PA during confinement periods, such as the COVID-19 pandemic, can be summarized from peer-reviewed publications”?

To provide relevant recommendations, this systematic review searched for recently published papers dealing with the current COVID-19 home-confinement situation, as well as journal articles dealing with previous pandemics or similar isolation situations.

## 2. Materials and Methods 

The systematic review was conducted and reported in accordance with the guidelines of the preferred reporting items for systematic reviews and meta-analysis—PRISMA [17].

### 2.1. Data Sources and Search Strategy

A comprehensive literature search was performed electronically on 16 May 2020 in the databases PubMed and Web of Science, considering all manuscripts published in English and German from inception. There were no restrictions for study design, setting, country, or time frame. The Medical Subject Headings (MeSH) thesaurus was used where appropriate. 

The search strategy was based on the psychological construct “situation”, as defined by the component’s “person”, “environment”, and “task”, which has been introduced for PA by Nitsch [18]. Brach et al. [19] applied this simple framework for describing how these components enable, but also constrain, individual mobility. For the present topic, we did not apply search terms for the “person” component, as practical recommendations are required for all population groups during pandemic situations. 

The “environment” component should be represented by search terms describing the home confinement situation. However, many different terms are used in public and academic articles, e.g., “social distancing”, “quarantine”, “self-isolation”. In order to avoid the risk of overlooking articles, we made use of the Medical Subject Headings (MeSH) thesaurus. This “thesaurus is a controlled and hierarchically-organized vocabulary produced by the National Library of Medicine” [20]. NIH staff assigns appropriate MeSH terms to each article included in PubMed. We used three “OR-combined” MeSH terms: “Patient isolation” (D010356), “social isolation” (D012934), and “pandemics” (D058873). We added the term “confinement” to identify all articles related to confinement phases. These four keywords were combined with the “task” component’s terms relating to PA (i.e., exercise, sports, and physical activity), as done by Hinrichs and Brach [21].

The final search strategy for PubMed was (“patient isolation” [MeSH Terms] OR “social isolation” [MeSH Terms] OR “confinement” [Title/Abstract] OR “pandemics” [MeSH Terms]) AND (“exercise” [Title/Abstract] OR “sports” [Title/Abstract] OR “physical activity” [Title/Abstract]).

The final search term for Web of Science (Topic search) was (“patient isolation” OR “social isolation” OR “pandemics” OR “confinement”) AND (“exercise” OR “sports” OR “physical activity” OR “physical fitness” OR “motor activity”).

### 2.2. Inclusion and Exclusion Criteria

To be included in the systematic review, each individual study was required to meet the following inclusion criteria: (i) Articles published in peer-reviewed journals in English or German, (ii) articles dealing with humans from any age categories (i.e., Child (0–12 years), Adolescent (13–18 years), Adult (19–59 years), and Senior Adult (60 years and above)), (iii) published before 16 May 2020. Exclusion rules were (i) studies written in languages other than English or German, (ii) animal studies, (iii) physiological studies, (iv) other conditions directly leading to social isolation (e.g., autism), and (v) articles without any connection to PA. 

### 2.3. Study Selection

The predefined search strategies yielded a preliminary pool of 1442 possible papers (585 in PubMed and 857 in the Web of Science) (see Figure 1), which were uploaded into the data management software Rayyan QCRI [22] to systematically screen these papers. Removal of duplicates resulted in a selection of 1206 published papers. Two researchers (EB and MB) independently screened titles and abstracts for eligibility against inclusion and exclusion criteria. After discussing and resolving 179 conflicting marks, a provisional list of 50 published studies went forward. Three full texts were not available at the time of submission for the present review. The full texts of 47 articles were carefully reviewed. Further, 16 articles were excluded (no focus on PA *n* = 14; only presenting study design *n* = 1, neither German or English language *n* =1). Therefore, 31 articles met our inclusion criteria.

### 2.4. Quality Assessment 

An overview of the six articles which explicitly deal with COVID-19 is provided in Table 1. The evidence level of the research topic and evidence base of PA recommendations from these studies are discriminated in Table 2.

This was necessary because PA was not always the core topic. With regard to PA recommendations, we assessed study type and rated the evidence base. The classification is (1) “Strong evidence” (scientific reviews of a large number of single studies), (2) “Medium evidence” (a small number of studies and individual reviews), and (3) “Weak evidence/not reached” (only relied on single or missing studies). This framework is used for the German national PA recommendations [23].

To assess the quality of all 31 integrated studies, we graded the papers into different evidence levels, which range from the best (level I) to the weakest (level VII) methodological quality (Table 2 and Table 3) [24]. The evidence levels are as follows: Level I refers to either a meta-analysis or even a systematic review of randomized controlled trials (RCTs) or evidence-based clinical practice guidelines, which rely on systematic reviews of RCTs or a minimum of three RCTs which report similar results; Level II is evidence obtained from a minimum of one RCT; Level III includes controlled trials without randomization; Level IV are cohort- or case-control studies; Level V is evidence from systematic reviews of qualitative and descriptive studies; Level VI refers to a single qualitative or descriptive study; Level VII gathers evidence from reports or expert committees, or opinions of authorities [24].

## 3. Results

This systematic review gathered relevant practical recommendations for performing PA during (i) the COVID-19 restrictions and (ii) similar pandemics and/or social isolation situations. Practical recommendations for staying physically active during home confinement gathered from COVID-19′s studies were summarized in Table 1. Main findings and recommendations gathered from studies dealing with other restrictive measures alongside the grading of evidence levels in these studies are visible in Table 3. 

### 3.1. Maintaining or Improving the Physical Activity Levels during the COVID-19 Pandemic

The formal types of the presented articles in Table 2 include four reviews, one article and one report paper. No empirical study during the current COVID-19 was found at the date of our systematic research. These six articles are directly referring to the COVID-19 pandemic, and discuss and recommend PA as an effective strategy to 

(i)mitigate the unwanted mental confinement consequences (i.e., poor sleep quality [25]; lower mental health [26])(ii)modulate the severity of the infection by supporting the immune system and improving metabolic health [27];(iii)mitigate the unwanted physical and social confinement consequences (i.e., poorer neuromuscular, cardiovascular, and metabolic health [28]; and social isolation [29]);promote an active lifestyle among obese children [30].

Altena et al. [25] discussed the relationship between PA and sleep quality during the current COVID-19 home confinement. The authors of this review reported that low levels of PA during the confined day negatively affected sleep quality. Given that major stress caused by the COVID-19 restrictions leads to sleep disruption, while PA is known to relieve stress and anxiety, regular PA was suggested as a preventive strategy against COVID-19 induced insomnia. Specifically, PA should be performed during the day, and if possible, regularly in daylight and not immediately before bed-time.

A narrative review by Chevance et al. [26] provides an overview of the mental health care in the COVID-19 epidemic in France, where restrictions did not allow for outdoor PA and social interaction beyond a single household. The authors conclude that maintaining whatever possible daily routine is essential during home confinement and specifically recommended to maintain PA. However, a detailed PA strategy during pandemics was not concretely described.

The review by Luzi et al. [27] addresses the relationship between obesity and influenza during the current COVID-19 and previous epidemics. This review suggests that PA interventions have the potential to modulate inflammation, support the immune system, and improve vaccination outcomes. It was also established that one of the determinants in the severity of influenza viral infections in obese patients is physical inactivity (sedentary behavior). Accordingly, PA was suggested as a protective strategy that improves immunological and metabolic health during pandemics.

In the same way, another article summarized impacts of sedentarism on neuromuscular, cardiovascular, and metabolic health due to the COVID-19 home confinement. In this paper, Narici et al. [28] reported that a sedentary lifestyle leads to a rapid loss of muscle mass, degenerative changes of neuromuscular system, reduced cardiorespiratory fitness, and an increased rate of mortality. Regarding the practical recommendations, authors of this paper suggested that, during the COVID-19 pandemic, PA strategy should include (i) high volume resistance exercise with a low to medium intensity, (ii) walking more than 5000 steps per day (with an advised option to start monitoring daily activity with a fitness tracker, e.g., smartwatch/apps), and (iii) conduct outdoors PA as often as possible (e.g., walking). 

The work of Pecanha et al. [29] described the social isolation and the increase in physical inactivity during the COVID-19 pandemic in the context of the global burden of cardiovascular disease. The assessment of smartwatch data revealed that the level of PA (steps per day) was reduced in the current pandemic, which accordingly increased cardiovascular risk factors. In this paper, authors suggest motivating elderly through supervised home-based programs in order to adhere to a low to medium exercise intensity program.

The brief, cutting-edge report by Pietrobelli et al. [30] investigates the effects of COVID-19 restrictions on lifestyle behaviors of obese children in Italy. The main findings from this study showed that the participants were less active (time spent in sports activities decreased by 2.30 ± 4.60 h/week) during the COVID-19 home confinement and had consequently gained weight, due to this unhealthy lifestyle. Previous studies revealed that the school environment reduces obesity risks, because it provides routine and structure in daily life [31]. However, unhealthy lifestyles during summer vacations lead to a higher body mass index [32]. The same phenomena were observed through the confinement period. Practical recommendations, such as providing adapted telemedicine lifestyle programs and guidance from practitioners in medicine, were suggested by Pietrobelli et al. [30] to encourage families to implement or maintain a healthy lifestyle in times of pandemics.

An overview of the methodological quality of the six studies which explicitly deal with COVID-19 is provided in Table 2. Three reviews [25,27,29] were classified to level II, because these articles refer to at least one RCT. In relation to the topic PA recommendation, two of these studies were graded to the evidence base medium [25,29] and one to weak [27]. The other review, article, and brief cutting-edge report [26,28,30] were ranged to level VI, because these articles refer to descriptive studies and reviews. With regard to the topic PA recommendation, the quality of two papers were classified to weak [26,30] and the other one to medium [28] evidence base.

### 3.2. Useful Results for an Active Lifestyle during Pandemics

The remaining 25 studies do not address the actual COVID-19 pandemic, but other social isolation situations alongside PA. Therefore, they may contribute to an active lifestyle, i.e., if recommendations can be transferred or used in other ways. For presentation purposes, we list five topics, which are useful for active lifestyle promotion, based on the author´s best assumptions. We then sorted the remaining articles to these topics (see Table 3).

The topic “Design of PA Programs” gathers general ideas for the development of future programs in times of isolation, home-confinement, and social distancing. Research reveals that programs to promote PA need to be adapted into the home environment. Therefore, connection to the internet is essential, which should be established in all communities. The government task is to prioritize internet telecommunications, thus preventing isolation and improving wellbeing [33]. To overcome barriers of conducting PA, the benefits of exercise should be explained. Furthermore, realistic exercise goals should be established [34]. PA interventions for the elderly should be encouraged in the afternoon, because at this time, most elderly demonstrate sedentary behavior [35].

The topic “Examples of Non-Digital Exercises and PA Programs” demonstrates currently available programs which could be adapted into the conditions present during a pandemic. One example is an indoor gardening program, by virtue of this activity’s positive effect on life satisfaction, social networking, and reduction in the perception of loneliness [36]. Another possibility is Tai Chi, because it can be adapted to a wide variety of age skill levels, and be modified for seated exercise or performed standing with or without walker support. Furthermore, the exercises can be performed individually or in a group [37].

The topic “Digital Health Technologies” focuses on technical solutions, which could be implemented individually for isolation, home-confinement, and social distancing. Being a member of a virtual fitness program increases PA level, and reduces loneliness and social isolation [38]. Exergames demonstrate beneficial effects for individuals with serious mental illnesses, for promoting physical health, improving self-efficacy, and an increase in social integration [39].

The topic “Space-Simulation Isolation” is a scenario which can be compared to the current pandemic. The research in this field shows that sedentary behavior and bed confinement should be overcome by performing exercise, such as resistance training, for the prevention of psycho-physiological changes [40,41]. 

The topic “Real-Life Isolation” addresses individuals, which experience confinement in general, not necessarily due to a pandemic. For example, homeless individuals are often isolated from the society. Nevertheless, their isolated experience can be overcome with the performance of activities, which leads to social interaction and better general health behavior [42].

An overview of the methodological quality of the 25 studies outlined above is provided in Table 3. One review [37] is classified at evidence level I, because this review includes systematic reviews, meta-analysis, and clinical practice guidelines. Four papers, including three reviews [40,48,54] and one RCT [49], were classified level II, because the reviews include at least three RCT´s and one gathered evidence based on their own RCT. Two quasi-experimental [36,38] and one controlled-study [55] were categorized level III. In these articles, the evidence is obtained from their own trial. One clinical review [44] is rated to level V, because it refers to systematic reviews and meta-analysis. The final 16 papers were classified to level VI, because these refer to their own qualitative study.

## 4. Discussion

The primary goal of this review was to provide practical recommendations to maintain an active lifestyle during the pandemic or other isolation period, through a systematic review of relevant articles. 

Thirty-one articles with relevant practical recommendations for maintaining or promoting an active lifestyle during pandemics were identified. Six of these explicitly addressed PA during the COVID-19 pandemic (see Table 1 and Table 2), and 25 articles discussed PA during similar confinement and/or isolation situations, with a view to transferring this knowledge to promote an active lifestyle during pandemics (see Table 3). 

Strong evidence shows that physical inactivity presents a major public health issue and suggests low-PA as major risk factor for decreased life expectancy and developing many non-communicable diseases (e.g., cardiovascular, cancer, and diabetes) [58]. In a pandemic situation including home confinement, it is difficult to meet the general WHO guidelines (150 min moderate to mild PA or 75 min intensive PA per week or combination of both) [10], because of pandemic-related restriction (e.g., citizens are not allowed to go outside) and/or reduced sports and other PA programs on offer (closed gyms, parks, centers).

In the following paragraphs, we will summarize the practical recommendations of the papers, which were included in the systematic review and combine them with additional literature (4.1). We also discuss the methodological quality of these articles (4.2). Additionally, the strengths and weaknesses of this review will be analyzed (4.3). Subsequently, future steps for a research program will be outlined (4.4).

### 4.1. Practical Recommendations

Main findings and recommendations from the included confinement studies revealed that the focus under pandemic restrictions should be moved toward home-based PA programs, which should be supervised in vulnerable individuals (e.g., elderly, individuals with diseases) [29]. This crisis-oriented program can be promoted through wearable technologies (activity trackers through smartphones/smartwatches), which have been widely suggested to provide an overview of daily activities and to help reduce inactivity and ensure improvements in cardiovascular parameters [28]. Consequently, it is important that individuals receive support and education in these new technologies. Governmental actions need to be taken to promote the role of PA as a health-care priority in general [29], but especially in times of pandemics where individuals with cardiovascular diseases are more vulnerable to serious infection. 

Specifically, it is suggested that regular daily routines and exercises in these home-based PA programs should be carried out in daylight [25]. When it is allowed, people should perform their exercises as often as possible outdoors, in nature while respecting distancing and hygiene precautions (e.g., walking or jogging) [28]. For monitoring steps per day, activity trackers should be used [28]. When monitoring is not possible, another option is to walk regular times per day, for example based upon WHO guidelines, a minimum 50 min in three days per week [10]. The intensity of all activities should be mild to moderate [27,59,60]. It can be monitored using the rating of perceived exertion (RPE) scale [61].

The RPE scale ranges from 0, which means rest, to 10, which relates to maximal exertion. By conducting mild to moderate activities, 3 is the limit, which represents moderate exertion [61].

Other exercises, such as strength training, should be performed additionally, with an intensity from low to medium [28]. Further, relaxation activities are relevant to reduce stress-level [25], which is often a reaction to the COVID-19 pandemic [62].

Included papers in the present review, also indicate that community-based programs have the potential to decrease feelings of isolation [45]. In the situation of home-confinement, exergames are a perfect innovation which should have the integrated function of a group-play modus [53]. Indeed, it is widely accepted that exergames have the potential to improve levels of PA [63] and increase the feeling of social belonging for older adults [64], including those with mental illness [65]. Smart technologies also have the potential to better manage and understand health conditions. With, or even without internet-access, Tai Chi is another possible exercise which can be adapted to different skills and different ages [37]. Dancing is another exercise option during home confinement, as it shows positive effects on brain health [66], fitness levels, and wellbeing [67], as well as a reduction in feelings of social isolation in patients with depression [68]. In cases where patients are in home confinement, telephone supervision or text messages should be established between health-care professionals and patients to provide personalized exercise programs [48]. 

Taken together, COVID-19 studies highlighted the maintenance of PA levels as a useful strategy to mitigate the psychosocial strain and the severity of infections during home confinement. These studies also revealed the importance of information and communications technology (ICT) to promote crisis-oriented PA programs.

Therefore, based on the included studies in Table 3, the authors of the present systematic review generated 5 topics which bundle information from examples of non-digital and digital PA programs and their design, space-simulation, and real-life isolation scenarios, which all present useful results which can be adopted in the pandemic restrictive conditions. These topics highlighted the potential of digital community exercise programs in motivating individuals and reducing their feeling of loneliness, even if they train from their home [45]. These topics reported that the first step of creating a PA program is to explain the benefits of the exercise, and to define realistic individual goals adapted to each group’s health status [34]. It is suggested that best case scenarios would integrate family members, because this leads to more motivation and an increased effort time for conducting PA [48]. Government actions should promote these PA programs as a health-care priority, especially for vulnerable individuals (e.g., elderly and patients with diseases) who may require a longer duration of isolation in order to avoid virus transmission. 

In order to create appropriate PA programs during pandemics, different practical materials for possible home-based exercises are recommended [29]. These exercises can be accessed from the World Health Organization (WHO), the American Heart Association (AHA), or the American College of Sports Medicine (ACSM). The training advice from the WHO is general. They advise 30 min PA per day for healthy adults, and 1 h per day for children. The proposed PA by WHO includes dancing to music, playing active video games, body-weight strength and balance training, and joining exercise classes online [69]. Additional practical ideas such as regular standing breaks [70], walking up and down the stairs [71], as well as the use of skipping ropes [72], are represented on the WHO exercise list, to ensure diversity in the content of the crisis-oriented PA programs. The AHA invites people to join a virtual workshop for different workouts (e.g., full-body workout, lower body tone workout, stretch workout). In addition, they offer useful exercises and explanations in their videos, such as a sofa stretch or chair dips [73]. Moreover, they compare activities in and around the house for times of pandemic restrictions with the following real-life-scenarios: “10 min of stretching is like walking the length of a football field”;“2.5 h of walking every week for a year is like walking across the state of Wyoming”;“30 min of singles tennis is like walking a 5K”;“1 h of dancing every week for a year is like walking from Chicago to Indianapolis”;“20 min of vacuuming is like walking one mile”;“30 min of grocery shopping every other week for a year is like walking a marathon” [74].

Through these comparisons, the reader can be motivated to be more active, which is an important precondition for conducting PA. They describe a home-based exercise program [75], with one challenge targeted at children [76]. Practical tips for the whole family are discussed, such as active chore cards, fitness during TV, active games (e.g., twister), or playing with pets [77].

Similarly, the home page of the ACSM provides different links to many blogs and home-based-exercise videos to maintain an active lifestyle during the COVID-19 home confinement [78]. Additionally, the reader can also find general information about recommended activities in a normal situation, omitting the pandemic restrictions [78]

### 4.2. Methodological Quality

The first impression when looking at the evidence levels of the COVID-19 studies is that half of the six demonstrate relative high quality (level II). The other three are of a lower quality (level VI). A more detailed analysis of these articles with regard to referred studies of the topic PA recommendation resulted in classifying three of them to medium and three of them to weak evidence base. It is important to point out that none of the authors have conducted an empirical study. This medium-to-low quality of the six studies which explicitly deal with COVID-19 is not surprising, because at the time of publication, the pandemic was novel. It was not possible to generate and conduct RCT’s or systematic reviews with meta-analysis in such a limited time. In relation to the 25 studies which address other pandemics and/or isolation situations, we identified that most of these were classified to the evidence level VI, which presents a low methodological quality. Eight papers were rated between I and III, and one was categorized to level V. This analysis shows that the results need to be interpreted with caution.

### 4.3. Strengths and Weaknesses

Strengths of this review are the timeliness of the subject, the multi-step research program, and a comprehensive coverage of the literature. High standards in search methodology were applied, e.g., using a theoretical framework and a well-established keyword system (MeSH) for generation of search strings, databases with peer reviewed articles, independent selection, and assessment of the evidence levels and the evidence base. On the other hand, so-called grey literature was not searched. Additionally, we need to acknowledge that the home confinement/social distancing/quarantine policies and practices are different across regions and countries. It therefore needs to be highlighted that this overview is a general one.

Substantially more articles were found in comparison to our early initial review [16], because included studies referred either to the actual COVID-19 pandemic (Table 1 and Table 2) or to other confinement situations (Table 3), and there were no restrictions of included article-types (e.g., reviews, reports, position-point papers, etc.). However, the usefulness and evidence of the recommendations is still limited. Largely, the recommendations are rather general, because they do not address specific age or target groups, and they are not categorized according to different social-distancing scenarios. Some practical recommendations of conducting PA are described in more detail for the different age-groups in the initial review. 

### 4.4. Further Research

Future research should adopt a higher methodological approach, in order to provide superior evidence-based knowledge, and should distinguish between recommendations and interventions for different age-groups (e.g., children, adults, older adults). It is also essential that research provides PA recommendations for individuals who were already affected by COVID-19, with analysis on when people should commence rehabilitation and activity. Further, consequences of closed fitness centers (e.g., mental health, potential weight gain) need to be analyzed. It is important for researchers to differentiate between different living and working conditions, as well as for different home confinement, social distancing policies and practices across regions and countries. This approach will be integrated into the wider research plan outlined in the introduction; the survey, the review, and the intervention branches of the research program. By increments, the research could progress from feasibility studies, formative evaluation of ongoing interventions, and case-studies to randomized controlled trials. Thus, transferable knowledge would be available immediately, while fostering the research base in parallel. 

## 5. Conclusions

The reduced PA levels, along with increased sedentary behavior, have been identified as serious concerns during home confinement resulting from the pandemic. This review demonstrates a small, but rapidly growing, body of evidence around practical recommendations to promote an active lifestyle during pandemics. Suggested PA exercises by WHO and AHA are helpful for the general population. However, more personalized, crisis-oriented PA programs targeting vulnerable populations, such as older adults and diseased individuals, need to be established. 

Filling such gaps of knowledge and application highlights the need to identify realistic goals for the PA programs, as well as the nature, volume, duration, and intensity of the recommended activities for each vulnerable sub-group. These PA programs can be assisted and delivered through ICT-solutions (e.g., fitness tracker, apps, online workout). The practical recommendations and applications of PA during confinement, discussed in the present review, need to be evaluated through other high methodological studies, and after that disseminated more broadly worldwide. Additionally, during pandemics, active lifestyles can be promoted through exergames which combine social belonging elements with home-based PA programs.

## Figures and Tables

**Figure 1 ijerph-17-06265-f001:**
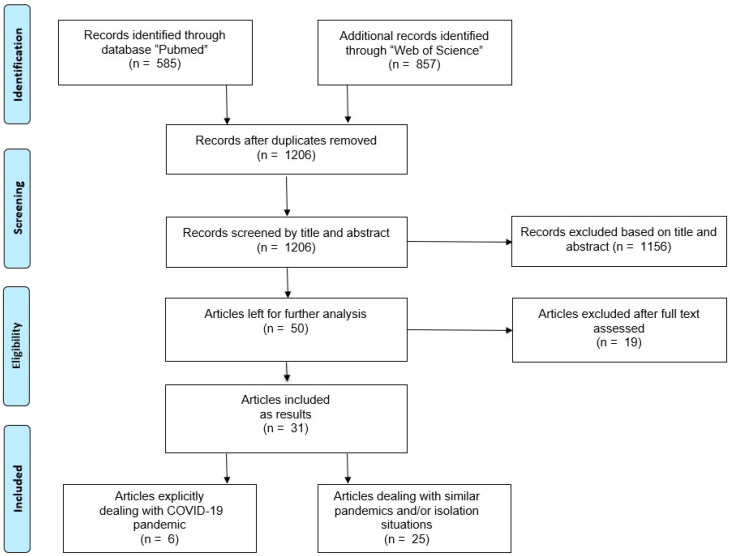
Results of literature search for physical activity during pandemics.

**Table 1 ijerph-17-06265-t001:** Practical Recommendations of studies for “Staying Active” during COVID-19 pandemic.

Autor andYear	Title	Methods	Results	Practical Recommendations
Altena et al. 2020 [25].	Dealing with sleep problems during home confinement due to the COVID-19 outbreak: Practical recommendations from a task force of the European CBT-I Academy	Summarizing information regarding the confinement and stress-sleep link and insomnia treatment	Stress and possible disruptions of social relationships can be avoided by dealing as with sleep problems.	Performing regular exercise Avoiding exercising directly before bed-time Being active in daylight Spending time in nature as often as possible (only if it is safe and permitted) Reducing stress with relaxation techniques Internet-based online programs for activity at home (e.g., for children)
Chevance et al. 2020 [26].	Ensuring mental health care during the SARS-CoV-2 epidemic in France: a narrative review	Analyzing the vulnerability of psychiatric patients to COVID-19 based on the existing literature, as well as a description of the procedures during the pandemic in the psychiatry setting.	Reorganizing of French psychiatry and simultaneously dealing with negative effects of home-isolation on mental health. A Major problem is a missing common voice for psychiatry in France to approach health authorities.	Maintaining regular daily routines Planning of activities Retrieving online WHO ^1^ recommendations Physical exercise as a general recommendation
Luzi & Radaelli 2020 [27].	Influenza and obesity: its odd relationship and the lessons for COVID-19 pandemic	Transferring findings about influenza and obesity for the visible risks of the COVID-19 pandemic.	Suggestions for weight loss in obese individuals with mild caloric restriction, including activators in drug treatment for obesity-diabetes patients and exercise (mild to moderate). Isolation should be longer than in normal weight individuals to reduce the risk of infection in this at-risk population.	Performing mild to moderate exercises
Narici et al. 2020 [28].	Impact of sedentarism due to the COVID-19 home confinement on neuromuscular, cardiovascular and metabolic health: Physiological and pathophysiological implications and recommendations for physical and nutritional countermeasures	Transferring results from models and real-life scenarios of inactivity to the COVID-19 home confinement.	Even a few days of inactivity and a sedentary lifestyle are enough to induce decreased aerobic capacity, insulin resistance, fat deposition, fiber denervation, neuromuscular junction damage, low-grade systemic inflammation, and muscle loss. Low to medium intensity high volume exercise in regular rhythms, combined with 15–25% caloric intake-reduction have the potential to prevent cardiovascular, metabolic, neuromuscular, and endocrine health.	Performing low to medium intensity resistive exercises Routines in exercising and daily exercise Following the rule that some physical activity is better than none (avoidance of sedentary lifestyle) Trying to take 5000 steps per day (use activity trackers) Conducting outside physical activities (e.g., walking, jogging)
Pecanha et al. 2020 [29].	Social isolation during the COVID-19 pandemic can increase physical inactivity and the global burden of cardiovascular disease	Providing an overview of interventions that could counteract a sedentary lifestyle and physical inactivity in cardiac patients and transferring of results to COVID-19.	Public agencies and health care professionals need to act together for the avoidance of premature death related to sedentary lifestyle. Even for patients with stable cardiovascular diseases home-based exercise programs with low to medium, to vigorous intensity exercises are effective and mostly safe. High-risk patients need special advice, which can be supported via telecommunication-services. Additionally, governmental actions need to be executed to reinforce physical activity promotion on a national level. The materials from scientific societies need to especially be disseminated among vulnerable groups (e.g., elderly, high risk patients).	Participating in home-based programs with low to medium intensity Motivate elderly in supervised home-based programs Using practical materials for home-based exercises from: WHO ^1^, AHA ^2^, ACSM ^3^
Pietrobelli et al. 2020 [30].	Effects of COVID-19 lockdown on lifestyle behaviors in children with obesity living in Verona, Italy: A Longitudinal Study	Analyzing pediatric individuals with obesity in the areas of activity, sleep, and diet behaviors before and during the COVID-19 home-confinement.	During the lockdown fruit intake increased.Sugary drink, red meat, and potato chip intakes increased as well. The time for sports participation decreased, sleep time and screen time increased. It can be assumed that, depending on duration, the pandemic may lead to negative effects on individual adiposity levels in children.	Accessing telemedicine lifestyle programs

^1^ World Health Organization. ^2^ The American Heart Association. ^3^ American College of Sports Medicine.

**Table 2 ijerph-17-06265-t002:** Methodological quality of studies for “Staying Active” during COVID-19 pandemic.

			With Regard to Author’s Topic		With Regard to PA Recommendations		
Article	Type Label (Journal)	Type (Our Judgement)	Methodology	Evidence Level	Types of Studies They Reference	Evidence Base	Comments with Regard to the Evidence Base
Altena et al. 2020 [25].	Review	Narrative review	Summarizing literature and making recommendations for the current situation based on theoretical basis.	Level II (paper refers to at least one RCT).	1 Review; 1 Systematic review and meta-analysis; 1 Clinical trial.	Medium	Proof of effectiveness based on individual reviews and one study. Limited recommendations can be made.
Chevance et al. 2020 [26].	Narrative review	Narrative review	Summarizing literature and transferring results to the current situation.	Level VI (paper refers mostly to descriptive studies and reviews).	1 Letter to the editor	Weak	Less evidence, only weak recommendation can be made based on the lack of theoretical background.
Luzi & Radaelli 2020 [27].	Narrative review	Narrative review	Summarizing literature and transferring results to the current situation.	Level II (paper refers to at least one well designed RCT).	1 Epidemiological study	Weak	Less evidence, only one weak recommendation can be made based on the lack of theoretical background.
Narici et al. 2020 [28].	Article	Narrative review	Summarizing literature and making recommendations for the current situation based on theoretical basis.	Level VI (paper refers mostly to descriptive studies and reviews).	1 Quantitative study; 1 Narrative review	Medium	Proof of effectiveness based on one review and one study. Limited recommendations can be made.
Pecanha et al. 2020 [29].	Review	Narrative review	Summarizing literature and making recommendations for the current situation based on theoretical basis.	Level II (paper refers to seven RCT´s).	5 RCT´s ^1^, 2 reviews, 1 crossover trial, 1 cohort study, 3 quantitative studies.	Medium	Proof of effectiveness based on two reviews and five different studies, including three different types. Some recommendations can be made.
Pietrobelli et al. 2020 [30].	Brief cutting-edge report	Qualitative study	Referring to a longitudinal study and adding new questions relevant to the current situation.	Level VI (paper refers to descriptive studies and a review).	Conclusion from their study.	Weak	Less evidence, only weak recommendation can be made based on the lack of theoretical background.

^1^ Randomized controlled trials.

**Table 3 ijerph-17-06265-t003:** Useful findings and recommendations to promote active lifestyle during pandemics.

Topic	Autor/Year	Title	Contribution for Possible Application in the Pandemic	Type Label (Journal)	Evidence Level
Design of PA ^1^ programs	Anderson et al. 2015 [43]	Fostering the Human-Animal Bond for Older Adults: Challenges and Opportunities	Pet ownership has the potential for reducing social isolation and an increase in PA.	Case study	Level VI (paper refers to descriptive and qualitative studies).
	Dunlap et al. 1999 [34]	Overcoming exercise barriers in older adults	For PA promotion, it is important to explain benefits of exercise, set realistic personal goals, control pain, treat chronic conditions, dispel misunderstandings.	Review article	Level VI (paper refers to descriptive and qualitative studies).
	Freedman et al. 2020 [44]	Social isolation and loneliness: the new geriatric giants Approach for primary care	Family doctors are positioned to identify socially isolated older adults and to initiate services.	Clinical review	Level V (paper is not a formal systematic review and does not provide a quantitative but a qualitative synthesis, of the field).
	Hwang et al. 2019 [45]	Loneliness and social isolation among older adults in a community exercise programme: a qualitative study	A community exercise program can motivate older adults to reduce their feelings of loneliness.	Qualitative study	Level VI (paper refers mostly to their qualitative study).
	Kearns & Whitley 2019 [33]	Associations of internet access with social integration, wellbeing and physical activity among adults in deprived communities: evidence from a household survey	Internet access has the potential to reduce social isolation, reinforce social inequalities, and generate better wellbeing.	Qualitative study	Level VI (paper refers mostly to their qualitative study).
	Leask et al. 2015 [35]	Exploring the context of sedentary behavior in older adults (what, where, why, when and with whom)	Interventions in the home environment, which focus on afternoon sitting time are necessary.	Cross-sectional exploratory study	Level VI (paper refers mostly to their qualitative study).
	Meinert et al. 2020 [46]	Agile Requirements Engineering and Software Planning for a Digital Health Platform to Engage the Effects of Isolation Caused by Social Distancing: case Study	Digital health platforms can combine societal and mental variables arising from social distancing measures.	Case-study	Level VI (paper refers mostly descriptive and qualitative studies).
	Nau et al. 2019 [47]	Enhancing Engagement with Socially Disadvantaged Older People in Organized Physical Activity Programs	A positive socio-cultural environment and the identification of activities of interest are preconditions for socially disadvantaged older people.	Qualitative study	Level VI (paper refers mostly to their qualitative study).
	Smith et al. 2017 [48]	The association between social support and physical activity in older adults: a systematic review	Social support, especially from family members, can lead to the performance of more leisure time PA.	Systematic review	Level II (review includes at least three RCTs ^2^).
Examples of non-digital exercises and PA-Programs	Hallisy 2018 [37]	Tai Chi Beyond Balance and Fall Prevention: Health Benefits and Its Potential Role in Combatting Social Isolation in the Aging Population	Tai Chi exercises can be adapted to a wide variety of skill levels and ages.	Review	Level I (review includes only systematic reviews, meta-analysis and clinical practice guidelines).
	Jansons et al. 2017 [49]	Gym-based exercise and home-based exercise with telephone support have similar outcomes when used as maintenance programs in adults with chronic health conditions: a randomised trial	Telephone support can have positive effects regarding exercise adherence.	RCT	Level II (evidence is obtained from their own RCT).
	Tse 2010 [36]	Therapeutic effects of an indoor gardening programme for older people living in nursing homes	An indoor gardening program can improve life satisfaction and social networking, and also decrease loneliness in older adults.	Quasi-experimental pre and posttest control group study	Level III (evidence is obtained mostly from their controlled trial.
	Murrock et al. 2016 [50]	Depression, Social Isolation, and the Lived Experience of Dancing in Disadvantaged Adults	Dancing should be considered for patients with depression and social isolation to develop a sense of belonging.	Qualitative study	Level VI (paper refers to their qualitative study).
	Sen et al. 2019 [51]	A Quality Mobility Program Reduces Elderly Social Isolation	A quality mobility program supervised by clinical professionals in a safe environment fosters sustained relationships that improve the quality of life and reduces social isolation in the elderly.	Case study	Level VI (paper refers mostly to their own qualitative study).
	Toepoel 2013 [52]	Ageing, Leisure, and Social Connectedness: How could Leisure Help Reduce Social Isolation of Older People?	Sports, cultural activities, voluntary work, holidays, reading books, shopping, and hobbies are found to be successful predictors for social connectedness of older people and friends support in participation in leisure activities. Therefore, local communities can use relationships and develop special programs for generating social connectedness.	Qualitative study	Level VI (paper refers to their own qualitative study).
Digital Health Technologies	Arlati et al. 2019 [53]	A Social Virtual Reality-Based Application for the Physical and Cognitive Training of the Elderly at Home	Virtual reality-based applications can implement the possibility to train with other users, which can reduce the risk of social isolation.	Development study	Level VI (paper refers mostly to descriptive studies).
	Brady et al. 2020 [38]	Reducing Isolation and Loneliness Through Membership in a Fitness Program for Older Adults: Implications for Health	Members of a digital fitness program increased their physical activity level and reduced social isolation and loneliness.	Cross-sectional, quasi-experimental study	Level III (evidence is obtained from their own controlled trial).
	Dobbins et al. 2020 [39]	Play provides social connection for older adults with serious mental illness: A grounded theory analysis of a 10-week exergame intervention	The group-play modus by exergames for older adults with mental illness can increase social belonging and a higher PA level.	Qualitative study	Level VI (paper refers mostly to their qualitative study).
	Morris et al. 2014 [54]	Smart technologies to enhance social connectedness in older people who live at home	Assistance to better manage and understand health conditions. Smart technologies (e.g., tailored internet programs) can generate improvements in aspects of social connectedness.	Systematic review	Level II (review includes six RCTs).
Space-Simulation Isolation	Alkner et al. 2003 [55]	Effects of strength training, using a gravity-independent exercise system, performed during 110 days of simulated space station confinement	Resistance training increases performance and maximal force output during long-term confinement.	Controlled study	Level III (evidence is obtained from their own controlled trial).
	Goswami 2017 [40]	Falls and Fall-Prevention in Older Individuals: Geriatrics Meets Spaceflight!	The comparison of astronauts, which spend large amounts of time in space and follow special exercise training, can be transferred to bed-confined older individuals.	Review	Level II (review includes three RCTs).
	Schneider et al. 2010 [41]	Exercise as a countermeasure to psycho-physiological deconditioning during long-term confinement	Exercises are useful to prevent psycho-physiological changes during confinement.	Qualitative study	Level VI (paper refers to their own qualitative study).
Real-Life Isolation	Abeln et al. 2015 [56]	Exercise in isolation—a countermeasure for electrocortical, mental and cognitive impairments	Regularly performed voluntary exercise supports subjective mental well-being of long-term isolated people.	Qualitative study	Level VI (paper refers to their own qualitative study).
	Malden et al. 2019 [42]	A theory-based evaluation of an intervention to promote positive health behaviors and reduce social isolation in people experiencing homelessness	After participating in the intervention, homeless individuals reported improvements in self-esteem, mental wellbeing, and social interaction. Additionally, their PA level and general health behavior had improved.	Qualitative study	Level VI (paper refers mostly to their qualitative study).
	Baidawi et al. 2016 [57]	Prison Experiences and Psychological Distress among Older Inmates	Psychological distress is connected with lower levels of exercise among older inmates.	Qualitative study	Level VI (paper refers to their own qualitative study).

^1^ Physical Activity. ^2^ Randomized controlled trial.

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
