# Peer review of "Practical Recommendations for Maintaining Active Lifestyle during the COVID-19 Pandemic: A Systematic Literature Review"

_ijerph, 2020, doi:10.3390/ijerph17176265_

Round 1
Reviewer 1 Report
Thank you for the opportunity to review this manuscript. This is an interesting and well-written study which provides insight physical activity during pandemics, including COVID19. It was nice to see recommendations for PA programs during pandemics (especially the recommendation to be connected to the internet in all communities!), although it feels like perhaps the authors could have done a broader search for published literature in this area. Table 2 was a great addition for practical info. Wonderful paper!
Below are some comments to consider:
Abstract:
-rewrite the second sentence. Of course a higher mortality rate would decrease cardiorespiratory fitness! Perhaps try “Consequences of inactivity, including a higher mortality rate and poorer general health and fitness, have been reported.”
-What is ICT? Spell out for the reader
Intro:
-Line 61: In this study, PA replaced sitting. You should note that in the text to be more clear
- Just a suggestion, but for the three branches, perhaps create a figure or diagram to show how these three flow, or are connected. Could be an interesting guide for future studies.
-Line 99: paper should be plural
Methods:
-Is there a reason only those two databases were searched? There were multiple others to review (Cochrane, CINAHL, Science Direct). If there is a reason for only those two, please state.
-Did you seek out the full-text of the three articles that you could initially not find full-text for? If they were excluded without being read, are you sure they would not have made it into the final set of publications? Please address that.
Line 152 – Perhaps add the figure caption “Figure A: Results of the physical activity during pandemics literature search”
Line 153 – Perhaps add the heading of “Results” here prior to the sentence
Line 231: Why did you choose these topics for Table 2? Was this based on the authors’ best assumptions as to what is applicable? If so, please state that. Perhaps also include a statement that Table 2 is not all-encompassing.
Discussion:
-Line 305: Not everyone has access or can afford an activity tracker. While it is a great recommendation, you might consider what are options for activity tracking in those less fortunate or with no access to them.
-One thing missing is evidence and/or discussion of PA in those affected with the illness of any pandemic. What might be PA recommendations for those individuals? I am not suggesting PA during their ICU/ventilation period; more so thinking of rehab and how soon they might be able to get active after recovering. Perhaps a minor point could be made about this in the future research section?
-Lastly, a topic not mentioned but perhaps related, is how to deal with gym/fitness centers closing. For individuals who relied on those for PA (whether due to equipment available or the need for social interaction while exercising, or whatever), are there unique needs for them to maintain activity during a home-confined pandemic? Further, if gyms remain closed for a long period of time, are there potential impacts that need to be considered (mental health, fitness, potential weight gain, etc) that results from the inability to be active at pre-pandemic levels? Just a thought that could be incorporated somewhere in the discussion.
Reviewer 2 Report
The current study provides valuable recommendations for policymakers to promote physical activities during the COVID-19 pandemic. However, some issues need to be addressed.
1) I could not see the relationship between the study background and research question. Please clarify.
2) The methodological quality of reviewed studies has not been evaluated. This shortage is a serious flaw. As results of the systematic review might directly contribute to evidence-based policy implementation, the poor methodological quality paper would result in an adverse effect on the public health system.
3) The search queries did not integrate several essential keywords, such as "social distancing", "quarantine", "self-isolation", etc. Therefore, there is a chance that not all related studies were included in the review.
4) The home confinement/social distancing/quarantine policies and practices are different across regions and countries. The current recommendations suggested by the study are too general and challenging to be applied. Categorizing recommendations according to different social-distancing scenarios might help.
5) The language needs to be substantially revised.
Based on these remarks, I recommend "rejection and resubmission".
Reviewer 3 Report
Congratulations for this review, it was really useful and I have only a few things to say:
Line 168 and 170: added final point at the end of the sentence.
Table 1: I think that it could be a good idea to include autor/title/method/result/conclusion (I think that the journal it not very important, you can see it in references section)
Discussion: I would start this section with the hypothesis that you expose in the introduction.
Round 2
Reviewer 2 Report
I would like to thank and congratulate on the hard and timely work that has been done by the authors in a short period of time. By adding quality refinement and assessment step, the authors have really improved the value and the methodological robustness of the systematic review. I, therefore, think the manuscript is now qualified for publication in the IJERPH.
One last remark, there are still several minor grammatical errors in the manuscript.